

# The study of the characteristics of the secondary flowering of *Cerasus subhirtella* 'Autumnalis'

Yanxia Xu[*], Liyan Du[*], Xuebin Song and Chunling Zhou

Qingdao Agricultural University, College of Landscape Architecture and Forestry, Qingdao, Shandong, China
[*] These authors contributed equally to this work.

## ABSTRACT

The short flowering period of ornamental cherry trees is the main factor limiting their use in gardens. Determining the secondary flowering characteristics of ornamental cherry trees is required to prolong their flowering period. In this experiment, *Cerasus subhirtella* 'Autumnalis' was used as the experimental material. The phenological differences in their annual growth cycle were observed using the BBCH coding system. The cooling requirements of the flower buds were evaluated by the chilling hours model (temperature between 0 and 7.2 °C) and the Utah model. The expression of the core gene involved in bud dormancy regulation *DAM* (dormancy-associated MADS-box) from the completion of flower bud differentiation in one year until the following year was measured by performing real-time fluorescence-based quantitative PCR. The results showed that the flowering duration of *C. subhirtella* 'Autumnalis' from November to December was longer than that of *C. yedoensis* 'Somei Yoshino', which was from March to April. The progress from seed bud-break to flower bud opening took about 10 days for *C. subhirtella* 'Autumnalis', while the same stage for *C. yedoensis* 'Somei Yoshino' took around 20 days. Additionally, the flower buds of *C. subhirtella* 'Autumnalis' needed only the chilling temperature unit of 54.08 to satisfy the chilling requirement, while *C. yedoensis* 'Somei Yoshino' required a chilling temperature unit of 596.75. After the completion of flower bud differentiation, during low-temperature storage, the expression of *DAM4* and *DAM5* genes first increased and then decreased, whereas, the expression of the DAM6 gene continued to decrease, and the expression of *DAM4*, *DAM5*, and *DAM6* in *C. yedoensis* 'Somei Yoshino' increased rapidly and was maintained at a high level. This showed that the upregulation of the expression of the *DAM4*, *DAM5*, and *DAM6* genes can inhibit the flower bud germination of Cherry Blossom. The relative expression of the *DAM* gene of *C. subhirtella* 'Autumnalis' was significantly lower than that of the *DAM* gene of *C. yedoensis* 'Somei Yoshino' from the end of October to the beginning of December, leading to its secondary flowering in autumn. These results might elucidate why the flower buds of *C. subhirtella* 'Autumnalis' can break their internal dormancy and bloom in the autumn and then again in the following year. Our findings might provide a reference for conducting further studies on the mechanisms of secondary flowering and bud dormancy in cherries.

Corresponding author
Chunling Zhou, 1456892420@qq.com

## INTRODUCTION

Cherry blossoms (*Cerasus* sp.) belong to the subfamily Prunus. of the Rosaceae family. They belong to a flowering tree species with beautiful flowers and have a high ornamental value. Cherry blossom is a widely used ornamental plant in urban green areas around the world, and it generally flowers in April. The flowering period is relatively short, which restricts its wide application as an ornamental plant in spring. Therefore, cultivating new cherry varieties with an improved flowering period is required. The cherry variety *C. subhirtella* 'Autumnalis' flowers in spring and October, which gives rise to a secondary flowering period. This flowering characteristic is very stable. Utilizing this secondary flowering variety can further develop the ornamental value of flowers in autumn, expand the effective ornamental period of cherry, and influence the cultivation of new cherry varieties with a favorable flowering period.

Many factors affect the stable secondary flowering of plants. The stages of flower bud development differ extensively based on the species and the season. For example, studies on flower bud differentiation of lilac (*Syringa oblata*) varieties with secondary flowering showed that the flower buds of the same plant might have two mechanisms of flower bud differentiation, which include the summer-autumn differentiation type and the primary differentiation type, in the same year (*Wang & Wang, 2009*). The cooling requirement is an important index to measure flower bud differentiation. During the dormancy period, the cooling requirements are not met, and the plants cannot pass through dormancy smoothly. This causes many problems that are not conducive to plant growth and development (*Alburquerque et al., 2008*; *Liu et al., 2017*; *Wang et al., 2003*). A study on the mechanism of secondary flowering in tree peonies (*Paeonia suffruticosa*) showed that most peony varieties with secondary flowering have low cooling requirements for flower bud-break, shallow dormancy, and a short period of flower bud expansion (*Chen, 2000*). Similar to peonies, cherry blossoms are plants that generally undergo genetic differentiation during summer and autumn. The flower buds of these plants require sufficient low-temperature storage at the dormant stage before they can germinate and open in the following year. However, *C. subhirtella* 'Autumnalis' can open from November to December of the same year after the flower bud differentiation is completed, indicating that the cooling requirements and the dormancy process have changed.

Determining the mechanism of secondary flowering by assessing the molecular mechanism of dormancy and dormancy release in plant buds and elucidating the endogenous process of inducing buds to stop growth is important (*Wang & Wang, 2009*; *Celgev & Su, 1955*). The gene related to dormancy in plants is called the *DAM* (dormancy-associated MADS-box) gene, which was first identified by Bielenberg et al. in the peach tree (*Prunus persica*) using forward genetics and map-based cloning (*Bielenberg et al., 2008*). Further studies on the *DAM* gene have shown that this gene can control plant dormancy and flower bud-break. For example, the bud-break of *Arabidopsis thaliana* seeds overexpressing the *DAM6* gene in cherry (*Cerasus pseudocerasus*) was strongly inhibited, and the heterologous expression of the *DAM1* and *DAM5* genes in cherry caused abnormal flower development in transgenic *Arabidopsis thaliana*. Most DAM genes can directly or

indirectly regulate bud dormancy in different species and inhibit bud growth, indicating that they are highly conserved (*Mei et al., 2018*; *Wan & Liu, 1979*). Some studies on peaches found that the expression levels of *DAM5* and *DMA6* were negatively correlated with flower bud-break, and the relative expression of these genes was upregulated in cold environments in the autumn. The expression of *DAM5* and *DAM6* in cold-demanding plant varieties was significantly higher than that in low-temperature cold-demanding varieties (*Yamane et al., 2011*). *Horvath et al. (2010)* showed that the *DAM* homolog *EeDAM* might be involved in the induction of endodormancy in the buds of leafy spurge (*Euphorbia esula* L.) (*Jiang et al., 2007*). *Jiménez, Reighard & Bielenberg (2010)* studied three peach tree varieties and found that the expression of *DAM* genes was higher before low-temperature storage. Under field conditions, the GC content of *DAM5* increased, followed by the decrease in the GC content of *DAM4* and *DAM6* genes after storage at low temperatures. When the expression of these genes in all varieties reached the lowest value, different varieties of peach trees had the same bud break capacity (*Jiménez, Reighard & Bielenberg, 2010*). A study on pear flower bud dormancy elucidated a molecular regulation mechanism involving the *DAM* gene. An increase in the relative expression of the *DAM* gene can induce plant buds to enter internal dormancy and maintain the state of intrabud dormancy by inhibiting the expression of the *FT2* gene (*Chen, 2000*). The *DAM* genes are most likely to be induced by low-temperature storage and variations in the internal dormancy of *C. subhirtella* 'Autumnalis'. The *DAM4*, *DAM5*, and *DAM6* genes are responsible for its secondary flowering. The expression level of these genes is the key indicator to elucidate the mechanism of secondary flowering of *C. subhirtella* 'Autumnalis'.

The secondary flowering phenology in plants has been recorded and reported across the world. The double-flowered plants include apple, pear, cherry blossom, lilac, plum (*Armeniaca mume*), Begonia (*Malus spectabilis*), Peony, *etc.* However, due to the lack of recognized phenological observation indicators, the comparison of phenological characteristics between different regions is hindered. The BBCH coding system (Biologische Bundesanstalt, Bundessortenamt, and Chemische Industrie) has provided unified quantitative standards for phenological observations and comparisons of plants in different regions and ways to solve the problem of inconsistent information on plant phenological developmental stages. The BBCH system is based on the Zadoks model and is jointly developed by BASF, Bayer, Ciba-Geigy, and Hoechst. It is a decimal coding system for describing the growth and development of herbaceous and woody plants and solves the problem of inconsistent description of plant phenological development scales. It is recognized and used by many countries and international organizations (*Zadoks, Chang & Konzak, 1974*; *Zhang et al., 2011*).

In this experiment, the double-flowered cultivar *Cerasus subhirtella* 'Autumnalis' (2n = 3x = 24) was used as the experimental material, and the once-flowering cultivar *Cerasus yedoensis* 'Somei Yoshino' (2n = 2x = 16) was used as the control. The two Sakura cultivars had the same parents. *Cerasus yedoensis* 'Somei Yoshino' is the most widely used and representative cherry variety in gardens (*Wang & Huang, 2001*). By determining the differences in phenological changes, cooling requirements, and DAM gene expression changes between the two varieties, the mechanism of secondary flowering of *Cerasus*

*subhirtella* 'Autumnalis' was elucidated. This study was conducted to provide new insights into the flowering of the genus Sakura, information on the bi-annual flowering of plants, and a theoretical basis for the related research on the application of cherry blossom in gardens.

## MATERIALS & METHODS

### Experimental materials

The experimental materials were collected from the campus of Qingdao Agricultural University (120°39′E, 36°32′N). The area has a warm temperate monsoon climate, an annual average temperature of 13.6 °C, a large daily temperature variation, and annual average precipitation of 467.4 mm. The cherry blossom varieties selected in the experiment were the double-flowered cultivar *C. subhirtella* 'Autumnalis' and the single-flowered cultivar *C. yedoensis* 'Somei Yoshino', which had the same parents. For each variety, three plants with similar growth rates and sizes were selected for the experiment.

### Experimental method
#### *Phenological observation*

From June 2019 to April 2021, the phenological observations and recordings were made based on the date of appearance of *C. subhirtella* 'Autumnalis' and *C. yedoensis* 'Somei Yoshino'. Observations were made every 3–5 days. When the phenological periods of these plant species overlapped and the plants grew rapidly, observations were made every day. We stopped monitoring the plants after they stopped growing in the winter. The specific phenological observation methods were developed based on the methods adopted by *Fadón, Herrero & Rodrigo (2015)* and *Wan & Liu (1979)*. The timing of the growth and plant phenology of each plant was recorded using the plant BBCH coding system. In this study, based on the BBCH coding system, the coding method for the secondary flowering of cherry blossoms was developed with repetitive phenological events in plants.

#### *Assessment of the chilling requirement*

The branches of *C. subhirtella* 'Autumnalis' and *C. yedoensis* 'Somei Yoshino' were collected in mid-October 2019. Each sampling set contained five long branches (≥15 cm) and five short branches (≤10 cm). The length and width of the buds on the branches were recorded every six days. The collected branches were maintained in an artificial climate room for subsequent cultivation. The artificial climate room was maintained at 24 °C/18 °C (day/night) and 62% relative humidity. The bottom of the branches was cut off obliquely. The water was changed every two days, and the base of the branches was cut 3∼4 mm to ensure complete water uptake. The data on field temperature and humidity were recorded with the COS-03 automatic temperature and humidity recorder, and the instrument was set to record data once every 10 min.

The bud-break data was statistically analyzed based on the method developed by Wanglirong et al. (*Wang et al., 2003*). For the bud-break rate, the grading of flower bud-break status was as follows: Level 1, no bud-break; Level 2, budding; Grade 3, bud tip exposed green; Grade 4, showing petals; Level 5, flowers open. The bud-break rate is the

weighted average value of flower buds at all levels, and the weight is the stage of the flower bud-break state. The grading of the chilling requirement was as follows: a bud-break rate $\geq 2.5$ indicated that the low temperature was passed; for 2.5 <bud-break rate $\leq 3.0$, the cooling demand was calculated based on the accumulated low-temperature hours in the field in this sampling period; for 3.0 <germination rate $\leq 3.5$, the chilling requirement was calculated based on the arithmetic mean of the field accumulated low-temperature hours in this period and the last sampling period; for bud-break rate >3.5, the chilling requirement of the last sampling period was considered to prevail. The 0 $\sim$7.2 °C model and the Utah model were used as the estimation model of the chilling requirement, and the formula to calculate the chilling requirement is as follows:

$$CH = \sum_{i=1}^{t} T, T = \begin{cases} 1 (0\,°C \leq T \leq 7.2\,°C) \\ 0 (else) \end{cases}$$

$$Utah = \sum_{i=1}^{t} = \begin{cases} T \leq 1.4 & 0 \\ 1.4 < T \leq 2.4 & 0.5 \\ 2.4 < T \leq 9.1 & 1 \\ 9.1 < T \leq 12.4 & 0.5 \\ 12.4 < T \leq 15.9 & 0 \\ 15.9 < T \leq 18.0 & -0.5 \\ T \geq 18.0 & -1. \end{cases}$$

### Extraction and identification of the DAM gene

Sampling was performed in mid-September 2020 when the flower bud differentiation of *C. subhirtella* 'Autumnalis' and *C. yedoensis* 'Somei Yoshino' was almost completed. Flower buds were collected every 15 days, 11 samples in total, and the branches that extended toward the sun were selected. The collected branches were placed in an ice box and transported to the laboratory. The collected flower buds were wrapped in tin foil, quickly frozen with liquid nitrogen, and stored in a refrigerator at −80 °C until analysis. The genes related to cooling requirements were searched in the NCBI (National Center for Biotechnology Information) database, and real-time quantitative PCR primers were designed based on the gene sequences reported in other studies (*Qiu et al., 2020*; *Chen, 2017*). The names of the synthetic primers and the upstream and downstream sequences used are presented in Table 1.

The total RNA of the samples was extracted using the Tiangen Polysaccharide Polyphenol Plant Total RNA extraction kit (DP441), and the obtained RNA solution was stored in a refrigerator at −80 °C to prevent degradation. The quality of the extracted RNA was visually assessed by agarose gel electrophoresis, and *cDNA* was synthesized using the PrimeScript™ 1st Strand *cDNA* Synthesis Kit (TaKaRa Bio Inc.). RSP3 was used as the internal reference gene in the real-time quantitative PCR (qPCR) technique, and the relative expression of the *DAM4*, *DAM5*, and *DAM6* genes in cherry blossoms was determined using the MA-6000 real-time quantitative PCR system.

**Table 1 Primer sequences.**

| Gene | Primer | Primer sequence | Purpose |
|------|--------|-----------------|---------|
| RSP3 | RSP3-F | TCAAGGTCAGGTAAGGGGGTC | Reference gene |
|      | RSP3-R | GTGAGGTGATTGTTAGTGGAAAGC | |
| DAM4 | DAM4-F | ATCAGATGGTGATGTTAAAGG | Full-length clone |
|      | DAM4-R | TGGAGGTAGCAGATTCAGAT | |
| DAM5 | DAM5-F | AATTGAATGATCAAGAGATTGAA | Full-length clone |
|      | DAM5-R | CAGCTCTTCCTTAGTTTCCATG | |
| DAM6 | DAM6-F | GAGTGAGATCATGTCACTGGAGAA | Full-length clone |
|      | DAM6-R | AGCTGGTAGAGGTGGCCATTGTG | |

## Data processing

The qPCR data were analyzed using the StepOne Software and then exported to Microsoft Excel and sorted into tables. The $2^{-\Delta\Delta Ct}$ method was used as a relative quantification strategy for the quantitative analysis of the data. All charts were made using Microsoft Excel.

## RESULTS

### Phenological observations

At the Qingdao Agricultural University, from June 2019 to April 2021, the annual growth and development cycles of *C. subhirtella* 'Autumnalis' and *C. yedoensis* 'Somei Yoshino' were observed, and the results of the phenological observations of two varieties of cherry blossoms were obtained. The phenological observations covered the whole growing season within the year. The main period of the development of *C. subhirtella* 'Autumnalis' and *C. yedoensis* 'Somei Yoshino' ranges from the bud development until all leaves are dropped, covering six of the 10 main growth stages based on the BBCH-scale system, including main growth stage 0: bud development, main growth stage 1: leaf development, main growth stage 3: shoot development, main growth stage 5: reproductive development, main growth stage 6: flower development, and main growth stage 9: defoliation initiation and dormancy of trees. The timing of the phenological stages is shown in Table 2. The expression method of the secondary flowering phenology of *C. subhirtella* 'Autumnalis' was developed by using the normal BBCH coding method for the first flowering (autumn flowering period) and adding the parentheses after the two-digit BBCH code of the first flowering to describe the second flowering (spring flowering period). Arabic numerals from 0 to 9 within the parentheses were then used to indicate the second repeat and complete flowering phenology.

After the flower buds sprouted in spring, the leaf buds of both the cherry blossom varieties started expanding in mid-March, and most leaves developed. These plants first bloomed and then developed leaves. Some leaves expanded fully from the end of March to the beginning of April. Most leaves unfolded fully in mid-April and late April, and the development lasted for about 30 days. The shoot development took longer, from the beginning of April to the end of August (about 140 days). Different branches of the same

**Table 2 The timing of phenological stages of two varieties of cherry blossom based on the BBCH coding system.**

| Principal growth stage | BBCH Code | Description of secondary growth stage | *C. subhirtella* 'Autumnalis' | *C. yedoensis* 'Someo-yoshino' |
|---|---|---|---|---|
| | 00 | Leaf buds closed and covered with scales | 03.09 | 03.12 |
| 0 Shoot development | 01 | Leaf buds begin to swell and the buds have scales with light edges | 03.12 | 03.14 |
| | 09 | Green leaf tips visible | 03.16 | 03.23 |
| | 10 | Green scales slightly open, leaves exposed | 03.22 | 03.28 |
| 1 Leaf development | 11 | Leaves unfold, developmental axis visible | 03.25 | 04.01 |
| | 12 | Leaves fully expanded | 04.03 | 04.07 |
| | 31 | Visible developmental bud axis | 04.04 | 04.09 |
| | 32 | Shoots 20% of the final length | 04.16 | 04.20 |
| | 33 | Shoots 30% of the final length | 04.22 | 04.25 |
| 3 Shoot development | 35 | Shoots 80% of the final length | 07.20 | 07.15 |
| | 39 | Shoots 90% of the final length | 08.28 | 08.30 |
| | 50 | Buds start to swell | 11.02 | \ |
| | 53 | Inflorescence surrounded by light green scales | 11.06 | \ |
| Flower development (autumn flowering period) | 54 | The pedicels start to elongate | 11.08 | \ |
| | 55 | First bloom | 11.12 | \ |
| | 57 | Half bloom | 11.13 | \ |
| | 59 | Full bloom | 11.13 | \ |
| | 50(0) | Dormant, inflorescence buds closed | 03.01 | 02.27 |
| 5 | 51(1) | Inflorescence buds swollen | 03.10 | 03.02 |
| | 53(3) | Inflorescence surrounded by light green scales | 03.16 | 03.12 |
| Flower development (spring flowering period) | 54(4) | The pedicels start to elongate | 03.19 | 03.17 |
| | 55(5) | First bloom | 03.20 | 03.22 |
| | 57(7) | Half bloom | 03.21 | 03.23 |
| | 59(9) | Full bloom | 03.22 | 03.23 |
| | 60 | Flowers bloom | 11.12 | |
| | 61 | Flowering begins | 11.12 | |

**Table 2** (*continued*)

| Principal growth stage | | BBCH Code | Description of secondary growth stage | *C. subhirtella* 'Autumnalis' | *C. yedoensis* 'Someo-yoshino' |
|---|---|---|---|---|---|
| 6 | Flower development (autumn flowering period) | 62 | 60% of flowers open | 11.18 | \ |
| | | 64 | 90% of flowers open | 11.24 | \ |
| | | 67 | Flowers begin to withe | 11.18 | \ |
| | | 69 | End of florescence | 12.14 | \ |
| | Flower development (spring flowering period) | 60(0) | Flowers bloom | 03.22 | 03.23 |
| | | 61(1) | Flowering begins | 03.24 | 03.23 |
| | | 62(2) | 20% flowers open | 03.25 | 03.24 |
| | | 64(4) | 60% flowers open | 03.27 | 03.25 |
| | | 65(5) | 90% flowers open | 04.01 | 03.28 |
| | | 67(7) | Flowers begin to withe | 03.30 | 03.30 |
| | | 69(9) | End of florescence | 04.12 | 04.10 |
| | 9 Leaves fall, dormancy begins | 91 | The twig growth is completed, and the leaves are still completely green | 08.28 | 09.08 |
| | | 92 | Blade starts to change color | 09.28 | 09.16 |
| | | 93 | Falling leaves begin | 10.17 | 10.22 |
| | | 95 | 50% of blades fall off | 11.08 | 11.10 |
| | | 97 | All leaves fallen | 11.28 | 11.24 |

plant were in different stages of development, and the shoots at the top of the crown required the longest time to develop. A few leaves of *C. yedoensis* 'Somei Yoshino' started changing color at the end of September, while those of *C. subhirtella* 'Autumnalis' changed color at the beginning of October. Both varieties started shedding leaves in mid-October, and all leaves were shed at the end of October. In deciduous plants, the dropping of leaves lasts for about 40 days, and plant dormancy is established during this time (*Westwood, 2009*).

After the differentiation of the flower buds of *C. yedoensis* 'Somei Yoshino' was completed, the plant entered internal dormancy. The flower buds started germinating in early March and developed in late March. The flower bud-break until the opening of the first bud took about 23 days. After the flower bud differentiation and the dormancy period of *C. subhirtella* 'Autumnalis' were completed, the plant quickly broke its inner dormancy in October, and the flower buds started germinating in early November. The ungerminated flower buds were forced into dormancy and started germinating in mid-March of the following year. The period from the bud-break of flower buds to the beginning of flowering took about 10 days. The required hours and suitable conditions were created for flower buds to completely open in the fall. *Cerasus subhirtella* 'Autumnalis' blooms twice a year. The first flowering occurs in autumn, while the second flowering occurs in spring, the duration of the two flowering periods is shown in Fig. 1. After the flower buds germinate and open in autumn, the pedicels, sepals, and stamens remain on the branches, until the bud breaks and opens in spring of the following year. The flower buds fall off at about the same time, the whole flowering process is shown in Fig. 2.
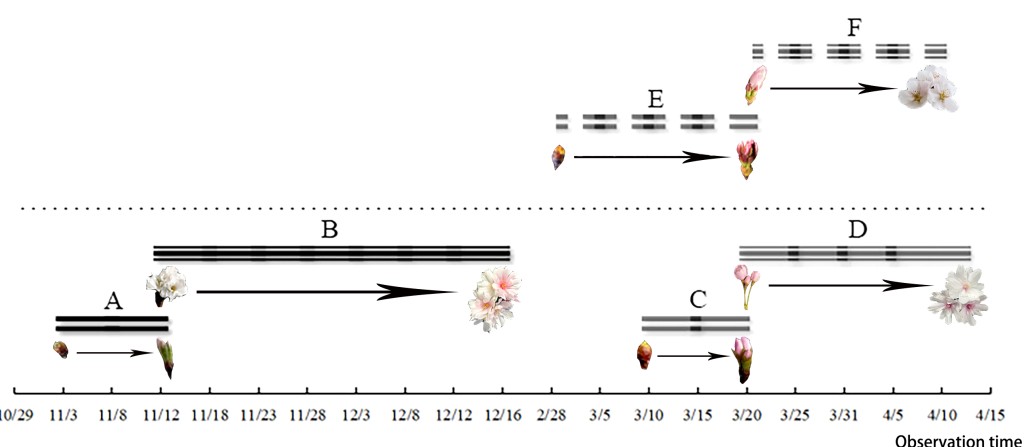

**Figure 1  Flower bud germination and the flowering duration of *C. subhirtella* 'Autumnalis' and *C. yedoensis* 'Somei yoshino'.** (A) The autumn-blooming period of flower buds of *C. subhirtella* 'Autumnalis', (B) the autumn-flowering period of *C. subhirtella* 'Autumnalis', (C) the spring-blooming period of *C. subhirtella* 'Autumnalis', (D) the spring-flowering period of 'October Sakura', (E) the period of the flower bud expansion of *C. yedoensis* 'Somei yoshino' in spring, (F) the spring-flowering period of *C. yedoensis* 'Somei yoshino'.

**Table 3  The chilling requirements of *C. subhirtella* 'Autumnalis' and *C. yedoensis* 'Somei yoshino'.**

| Variety | 0~7.2 °C model | | Utah model | |
|---|---|---|---|---|
| | Starting and ending time | Results of cooling requirement | Starting and ending time | Results of cooling requirement |
| C. subhirtella 'Autumnalis' | 11\20- | 0 C H | 10\22–11\7 | 54.08 C U |
| C. yedoensis 'Someo-yoshino' | 11\20–1\10 | 523.00 C H | 10\22–1\10 | 596.75 C U |

## The chilling requirement of flower buds

The results (Table 3) of the Utah model to calculate the chilling requirement showed that the minimum low-temperature storage of the flower buds of *C. subhirtella* 'Autumnalis' was 54.08 C U, and the minimum chilling requirement of the flower buds of *C. yedoensis* 'Somei Yoshino' was 596.75 C U. The results of the 0–7.2 °C model to calculate the chilling requirement showed that the flower buds of *C. subhirtella* 'Autumnalis' germinated without low-temperature storage, and the lowest chilling requirement of the flower buds of *C. yedoensis* 'Somei Yoshino' was 523.00 C H. *Darbyshire et al. (2011)* divided the cooling requirements of sweet cherries into four intervals based on the level of cooling requirement, and the chilling requirement of the flower buds of *C. subhirtella* 'Autumnalis' was lower than the standard with the minimum intervals of chilling. Overall, the two chilling requirement estimation models showed that the chilling requirement for the flower bud-break of *C. subhirtella* 'Autumnalis' was significantly lower than that of *C. yedoensis* 'Somei Yoshino'.

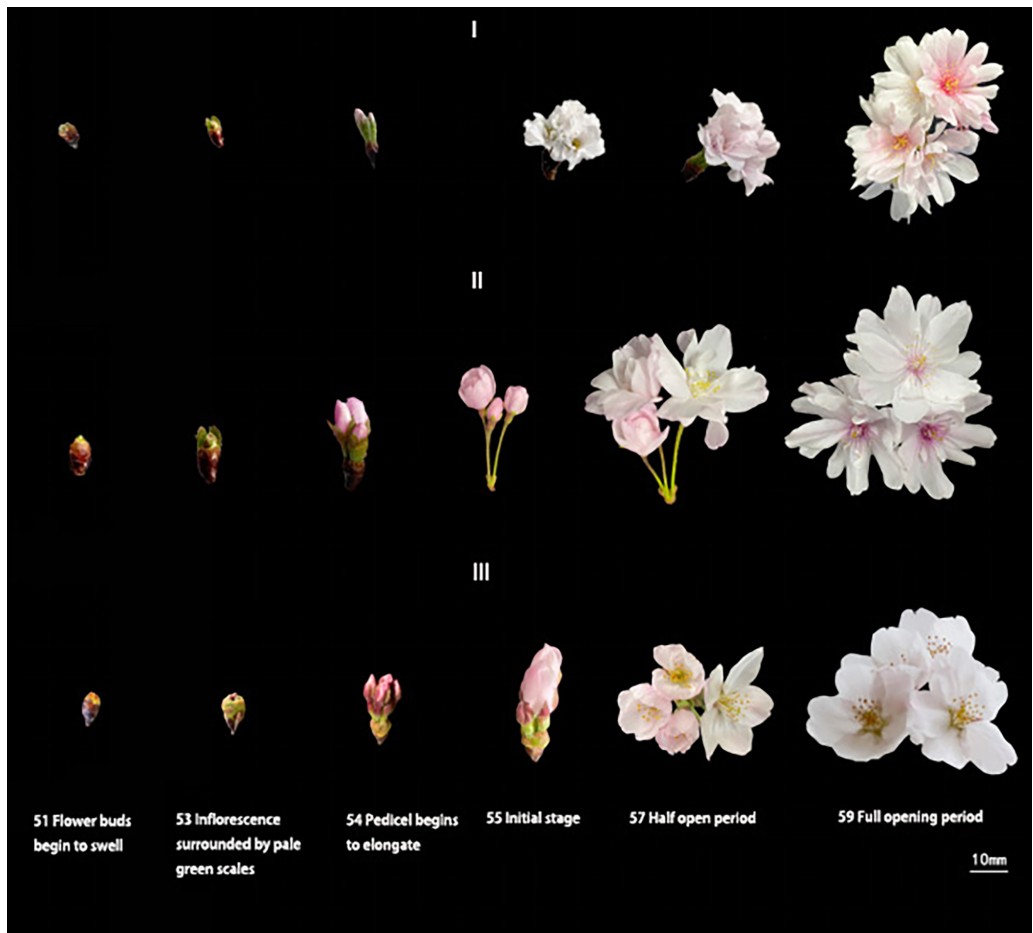

**Figure 2** **Flower bud germination and opening of *C. subhirtella* 'Autumnalis' and *C. yedoensis* 'Somei yoshino'.** (I) The process of flower bud germination and opening of *C. subhirtella* 'Autumnalis' in autumn, (II) the process of flower bud germination and opening of *C. subhirtella* 'Autumnalis' in spring, (III) the process of flower bud germination and opening of *C. yedoensis* 'Somei yoshino' in spring, (S1) 51, The flower bud-swelling phase, (S2) 53, Inflorescence enclosed by pale green scales, (S3) 54, Pedicel begins to elongate, (S4) 55, Initial stage, (S5) 57, the half-opening period, (S6) 59, The full-opening period.

## Expression of the *DAM* gene
### Changes in the relative expression of DAM4

The *DAM* gene triggers dormancy and inhibits flower bud-break in plants. In this study, after the plant entered dormancy, the expression of the *DAM* gene was upregulated rapidly, and then, the expression of the *DAM* gene was downregulated with storage at a low temperature. The relative expression of the *DAM4* gene firstly increased and then decreased at the dormant stage of *C. subhirtella* 'Autumnalis' and *C. yedoensis* 'Somei Yoshino' (Fig. 3), but the timing and amplitude of the relative expression of the *DAM4* gene in these two varieties differed significantly. Moreover, the relative expression of the *DAM4* gene in the flower buds of *C. yedoensis*' Somei Yoshino' and *C. subhirtella* 'Autumnalis' was extremely low in September. After the flower bud differentiation was completed in mid-October, the relative expression of the *DAM4* gene in the flower buds of *C. yedoensis*

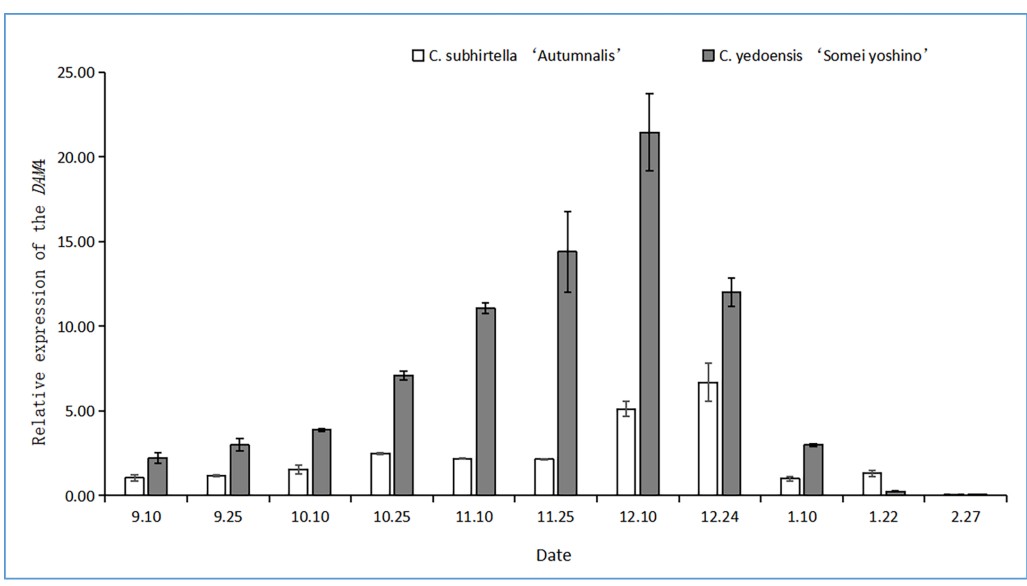

**Figure 3** Relative expression of the DAM4 gene in flower buds of *C. yedoensis* 'Somei yoshino' and *C. subhirtella* 'Autumnalis'.

'Somei Yoshino' increased till December. The relative expression of *DAM4* reached the highest value in mid-December. In early January of the following year, the flower buds of *C. yedoensis* 'Somei Yoshino' accumulated cold energy. The relative expression of the *DAM4* gene decreased rapidly to the same level as that before entering dormancy by the end of January. The relative expression of the *DAM4* gene in the flower buds of *C. subhirtella* 'Autumnalis', however, increased slightly in mid-October and decreased to the same level as that before entering inner dormancy during autumn-flowering. The relative expression of the *DAM4* gene was significantly upregulated in mid-December at the end of the autumn-flowering period. In *C. subhirtella* 'Autumnalis', the upregulation occurred within a shorter period and at a lower amplitude than that in *C. yedoensis* 'Somei Yoshino'.

### Changes in the Relative Expression of DAM5

The functional analysis of MADS-box genes related to flower bud dormancy in the Chinese cherry in other studies showed that the *DAM4*, *DAM5*, and *DAM6* genes might be involved in flower bud dormancy and dormancy release (*Shao, 2016*). The relative expression of the *DAM5* gene in the flower buds of *C. yedoensis* 'Somei Yoshino' first increased and then decreased during dormancy (Fig. 4). However, the relative expression of the *DAM5* gene in the flower buds of *C. subhirtella* 'Autumnalis' showed a fluctuating trend, *i.e.,* it was first upregulated and then downregulated, followed by the same trend. After the differentiation of the flower bud of *C. yedoensis* 'Somei Yoshino' was completed in mid-September, the relative expression of the *DAM5* gene increased rapidly, and in *C. subhirtella* 'Autumnalis', it increased at a lower rate compared to that in *C. yedoensis* 'Somei Yoshino' in the same period. From October to November, the relative expression of *DAM5* in *C. yedoensis* 'Somei Yoshino' was high, while in *C. subhirtella* 'Autumnalis', it was first upregulated and then

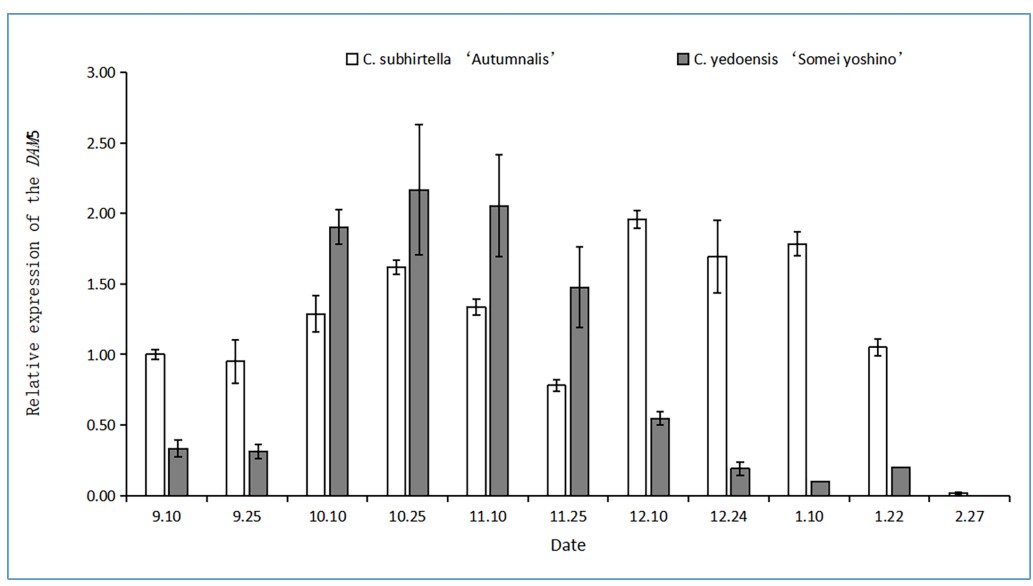

**Figure 4** Relative expression of the DAM5 gene in flower buds of *C. yedoensis* 'Somei yoshino' and *C. subhirtella* 'Autumnalis'.

downregulated. The low-temperature storage caused the inner dormancy to break, and the relative expression of *DAM5* was the lowest during the autumn-flowering period. In mid-December, when *C. subhirtella* 'Autumnalis' entered dormancy, the relative expression of the *DAM5* gene in flower buds increased, and the ungerminated flower buds could not continue to germinate. From mid-December to early January, the relative expression of the *DAM5* gene in the flower buds of *C. subhirtella* 'Autumnalis' was maintained at a high level, and then, it decreased till the end of January. However, in the flower buds of *C. yedoensis* 'Somei Yoshino', the relative expression of the *DAM5* gene remained at a high level from mid-December to early January. It decreased gradually to the lowest level at this time, and the flower buds of *C. yedoensis* 'Somei Yoshino' germinated and opened. These plants generally could not open because the external environmental conditions were not suitable for their growth. At the end of February, when the flower buds began germinating, the relative expression of *DAM5* in *C. yedoensis* 'Somei Yoshino' and *C. subhirtella* 'Autumnalis' reached the lowest levels.

### Changes in the relative expression of DAM6

The relative expression of the *DAM6* gene in the flower buds of *C. yedoensis* 'Somei Yoshino' was first upregulated and then downregulated during dormancy (Fig. 5). It first decreased, then increased, and finally, decreased again. During low-temperature storage, the relative expression of the *DAM6* gene was rapidly downregulated and was the lowest during the autumn-flowering period of *C. subhirtella* 'Autumnalis'; however, it increased after the autumn-flowering period was over. In contrast, in the flower buds of *C. yedoensis*

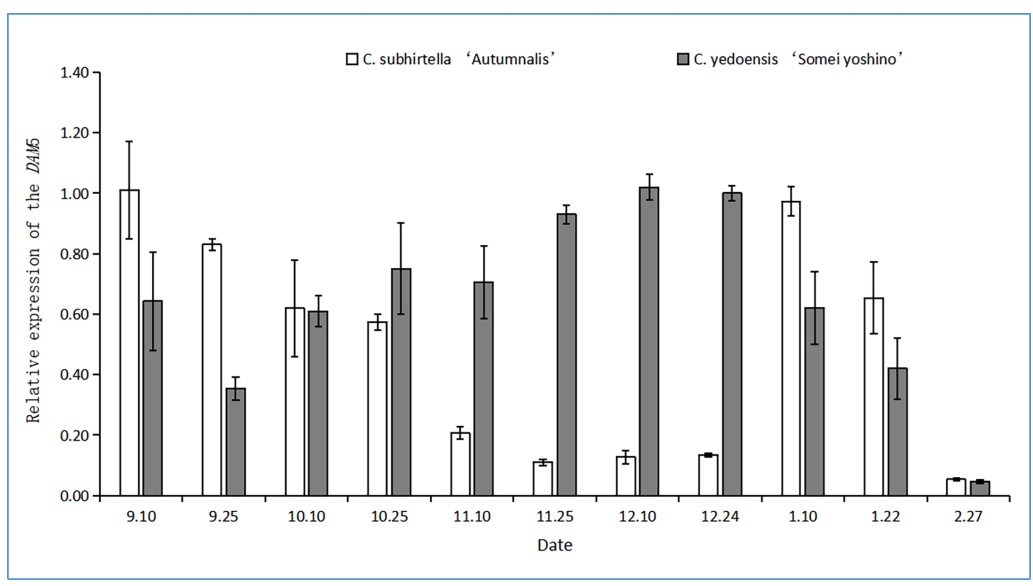

**Figure 5** Relative expression of the DAM6 gene in flower buds of *C. yedoensis* 'Somei yoshino' and *C. subhirtella* 'Autumnalis'.

'Somei Yoshino', the relative expression of the *DAM6* gene remained at a high level during the same period.

## DISCUSSION

### Differences in phenological stages

In Qingdao, the autumn-flowering period of *C. subhirtella* 'Autumnalis' is from early November to mid-December, with a flowering duration of about 35 days, whereas, the spring-flowering period is from the end of March to mid-April, with a flowering duration of about 20 days. The spring-flowering period of *C. yedoensis* 'Somei Yoshino' is from mid-late March to mid-April, and the flowering duration is about 16 days. The spring-flowering period of *C. subhirtella* 'Autumnalis' starts slightly earlier than that of *C. yedoensis* 'Somei Yoshino', and the duration of the spring-flowering period of *C. yedoensis* 'Somei Yoshino' is slightly shorter than that of *C. subhirtella* 'Autumnalis', but the flowering is more concentrated. The flowers of *C. yedoensis* 'Somei Yoshino' are in full bloom, and they have more flowers than *C. subhirtella* 'Autumnalis'. By comparing the two flowering periods of *C. subhirtella* 'Autumnalis' within one year, we found that the autumn-flowering period lasts longer than the spring-flowering period in this variety.

### The chilling requirement of *C. subhirtella* 'Autumnalis' was significantly lower than that of *C. yedoensis* 'Somei Yoshino'

*Cerasus subhirtella* 'Autumnalis' can bloom in autumn, but the inconsistent flowering is closely related to the lower chilling requirement and the characteristics of flower bud differentiation. The time of initiation and the rate of flower bud differentiation in the same *C. subhirtella* 'Autumnalis' plant differ (*Zheng, 2000*). The low-temperature storage

first occurs in those flower buds that first complete the differentiation process. Due to a significant decrease in chilling hours, the flower buds can gradually complete the low-temperature cycle in autumn. Accumulation, breaking internal dormancy, bud-break, and opening occur before the temperature drop triggers the flower buds to enter forced dormancy, which causes *C. subhirtella* 'Autumnalis' to bloom in autumn. However, flowering is not concentrated, and only a few flowers occur in full bloom. The physiological mechanism of autumn-flowering of *C. subhirtella* 'Autumnalis' is similar to that of peonies (*Wang, 2007*).

### Relative expression of *DAM* Genes

The relative expression of the *DAM4* gene in *C. yedoensis* 'Somei Yoshino' and 'Yueyueying' was considerably higher than that of *DAM5* and *DAM6* at the dormant stage. These results were similar to those found by *Chen (2017)* in a study on sweet cherries. The *DAM4* gene plays a dominant role in the bud dormancy process of cherry blossoms. The results of the experiments showed that the relative expression of the *DAM4* gene in the flower buds of *C. subhirtella* 'Autumnalis' with low cooling requirements during dormancy was significantly downregulated compared to the relative expression of the *DAM4* gene in the flower buds of *C. yedoensis* 'Somei Yoshino'. These results were similar to those reported by *Ubi et al. (2010)*. The results of the study on the *DAM* genes in the dormant stage were similar. The high-chilling requirement cultivars had a high level of *DAM* gene expression in the dormant stage. With the release of dormancy, the expression was downregulated, and the expression of the low-chilling requirement cultivars also decreased significantly. During the autumn-flowering period, the relative expression of the *DAM5* gene in *C. subhirtella* 'Autumnalis' was significantly lower than that in *C. yedoensis* 'Somei Yoshino'. The *DAM5* gene inhibits the bud-break of flower buds. The bud-break rate of buds was negatively correlated with the relative expression of the *DAM5* gene, and the expression level in the flowers was extremely low, indicating that the *DAM5* gene could inhibit bud-break. Another study (*Balogh et al., 2019*) also showed that *DAM5* and *DAM6* can delay the dormancy release and flowering time and inhibit flowering in plants.

The *DAM6* gene may play an important regulatory role in the process of bud dormancy and dormancy release in cherries. The relative expression of the *DAM6* gene first increased and then decreased with low-temperature storage (*Mei et al., 2018*). Before entering the internal dormancy period, the relative expression of *DAM6* in the flower buds of *C. yedoensis* 'Somei Yoshino' and *C. subhirtella* 'Autumnalis' was higher. Different *DAM* genes can inhibit flower bud-break at different times, and the *DAM6* gene mainly influences processes occurring before the completion of flower bud differentiation.

## CONCLUSIONS AND OUTLOOK

The flower bud differentiation of *C. subhirtella* 'Autumnalis' increased, and the period from flower bud-break to flowering decreased, providing optimum conditions for buds to open in autumn. The buds of *C. subhirtella* 'Autumnalis' had a lower cooling requirement, and the *DAM* gene was expressed in the buds flowering in autumn. As low-temperature storage decreased rapidly, flower buds could break their internal dormancy and continue to develop

before entering eco-dormancy, which is an important reason for the autumn-flowering of *C. subhirtella* 'Autumnalis'. Studies on the mechanism of the secondary flowering of *C. subhirtella* 'Autumnalis' are limited, and the experimental anatomical data on the structure of *C. subhirtella* 'Autumnalis' during the critical period of secondary flowering are insufficient. The mechanism related to the effects of the *DAM4*, *DAM5*, and *DAM6* genes, and the regulatory mechanism of the internal dormancy and release of internal dormancy in the flower buds of *C. subhirtella* 'Autumnalis' need further investigation.

## ACKNOWLEDGEMENTS

Thanks to the central laboratory for providing equipment support during the experiment period, and thanks to all the teachers for their suggestions and guidance on the paper.

### Funding

This work was supported by the National Natural Science Foundation of China (No. 3180020316) and the National Key Wild Protection in Some Areas of Qingdao (2313211). The funders had no role in study design, data collection and analysis, decision to publish, or preparation of the manuscript.

### Grant Disclosures

The following grant information was disclosed by the authors:
National Natural Science Foundation of China: 3180020316.
National Key Wild Protection in Some Areas of Qingdao: 2313211.

### Competing Interests

The authors declare there are no competing interests.

### Author Contributions

- Yanxia Xu conceived and designed the experiments, performed the experiments, analyzed the data, prepared figures and/or tables, authored or reviewed drafts of the article, and approved the final draft.
- Liyan Du conceived and designed the experiments, performed the experiments, analyzed the data, prepared figures and/or tables, authored or reviewed drafts of the article, and approved the final draft.
- Xuebin Song conceived and designed the experiments, authored or reviewed drafts of the article, and approved the final draft.
- Chunling Zhou conceived and designed the experiments, authored or reviewed drafts of the article, and approved the final draft.

### Data Availability

The raw measurements are available in the Supplemental Files.

## Supplemental Information

Supplemental information for this article can be found online at http://dx.doi.org/10.7717/peerj.14655#supplemental-information.

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
