# Peer review of "The study of the characteristics of the secondary flowering of Cerasus subhirtella ‘Autumnalis’"

_PeerJ, doi:10.7717/peerj.14655_

## Round 0.1 · original submission · Major Revisions

· Academic Editor

Major Revisions

Dear Sir

As you can see, the manuscript needs major revision, with many important comments from both reviewers. Kindly revise your manuscript accordingly

Reviewer 1 ·

Basic reporting

Secondary flowering is of great significance to ornamental plants. This study offers some fundamental information about this phenomenon in cherry blossoms with contrasting flowering patterns. Though the data is basic and descriptive, it provides a theoretical basis for the rational development and utilization of cherry blossom resources.

The description logic of the results is not clear in the Abstract, and the enlightenment of the existing results to the relevant research is not sorted out as well
(1) Autumnal flowering time of C. subhirtella 'Autumnalis' is longer than that of C. yedoensis 'Somei Yoshino' in spring. Why did you compare the spring flowering time of one species with the autumn flowering time of another?
(2) What's the purpose to compare the time needed from seed germination to flower bud opening in the two cherry species?
(3) The description of DAM gene expression changes is vague. It is not clear how different DAM genes change in the two cherry species. In addition, what is 'Ranjingjiye'?

Manuscript writing:
There are many grammar problems, please polish the language.
The Introduction section is too wordy and illogical, which needs to be reorganized.

For example:
Line 51 The main factors used in …? This sentence is not complete.
Line 52-55 This sentence is too lengthy.
Line 67 BBCH coding system is an important strategy in this study, so a further description and detailed application of it should be provided in the manuscript.
Line 71 Since secondary flowering is the main focus of this manuscript, its definition and types should be introduced first.
Line 79 This sentence is confusing. How can unstable secondary flowering be closely related to genetic material?
Line 90-91 genetic differentiation or flower bud differentiation?

Many similar problems should be checked carefully in the full text.
The second and third paragraphs of the Introduction section involved too much content. It is suggested that they could be divided into several paragraphs according to logic.

Formatting
1、Please check the use of Latin names (such as Line 111, the Latin name of the peach tree was wrong)and ensure they were in italic.
2、 Please check the format of citations in the text and reference lists (such as Line 111, the year of a citation from Bielenbery et al. was missing)
3、Please ensure the first words in all sentences were in uppercase (Line 169)
4、Please check the display of units or symbols in the formulas
5、Please give the full gene name when it appeared for the first time.

Figures
Fig.1 What’s the difference between the blooming period and flowering period?
Fig.2~4 Please add the meanings of bar and line charts in figure legends and perform a difference significance analysis.
Fig.5 It might be better to integrate these photos in Fig.1.

Experimental design

1、The number of each variety as experimental materials is too small.
2、Please introduce the application of the BBCH coding system in this study with more details.
3、Please add detailed environmental information about the artificial climate room (Line 171) and the manufacturer and model of the equipment (Line 174) used in this study.

Validity of the findings

1、The starting and ending dates of phenological observation were not clear.
2、 The process of flower bud differentiation is important in the secondary flowering of plants. Why did you take samples until the flower bud differentiation was complete and how many times did you take samples?
3、The description of the results should be more concise.
4、Please perform difference significance analyses for the data obtained.

Reviewer 2 ·

Basic reporting

The English of the document should be revised by a native speaker. Furthermore, ambiguous references exist for key concepts of the manuscript (e.g. chilling temperature unit or cooling requirements instead of the widely used “chilling requirements”).
Also, it may be confusing for the reader the reference to germination (in the abstract in the main text). Germination is usually used for seeds. Do the authors try to refer to bud-break? Please change this term throughout the manuscript.
The introduction may need to be restructured and revised. There is no a clear flow of ideas and the need of the current study has not been stated.
Some other comments on the Introduction:
Prunoideae usually refers to the subfamily within the Rosaceae family. The genus would be Prunus.
The first paragraph of the introduction needs a complete rephrasing. Also, the way of using of references is not correct (e.g. “The main factors used in (Wang & Huang 2001).”). The authors keep citing the same article (Wang & Huang 2001). It may be better to have different sources for such a long paragraph.
The description of the species is missing the genome size, the ploidy and information on the phylogeny of the two species studied in this article. This information is crucial for all the analysis downstream.
L63. Which country?
L78-L81. Which plants? The phrase is vague. Please specify which type of plants you are referring to.
L101. The authors refer to the Utah model citing: Zhuang et al., 2012. The Utah Model was developed in 1974 by Richardson et al. Please modify accordingly across the document.
L111. Bielenberg et al.,(). Please revise the references thorough the manuscript.
L118-L121. Vernalization and bud dormancy are essentially different processes. The authors may want to rephrase this sentence.
L126. DAM6
L178. The formula cannot be properly read. Please modify accordingly.
L258. Cooling requirements is not the term usually used. Please modify to chilling requirements throughout the manuscript. Also, it is suggested to use the Dynamic model for chilling requirement (see Fishman et al., 1987). I would recommend to add Chill Portions unit from the Dynamic model.

The figures require further improvement. In Figure 1 the dotted line is not explained. Figures 2-4 have a different nomenclature for the months than Figure 1. Figure 5 show text that is overlapping and has different font size.

Raw data is presented, but please, do label the data in English. The excel tab (and in other parts) is not in English.

Experimental design

The research presented in this manuscript does not fit within the Scope of the Journal. The manuscript does not fit the scientific and methodological soundness required to publish in PeerJ.

The research question is not well defined in the introduction.

Methods are not sufficiently described to replicate the analysis. For example, in L168-178 no information is given about:
1) the frequency of the sampling (there was just one sampling date?),
2) temperature of the chamber,
3) the method (a short description of “Wanglirong et al. (Wang et al. 2003).” is recommended to be added).
4) the choice and validity of the reference genes for the studied species,
5) the number and characteristics of the DAM genes in the studied species. Are the same as in the species cited in the study?

Validity of the findings

The authors may need to provide more details in the manuscript for the correct evaluation of the validity of the findings.

Additional comments

The authors may need to work thorough several aspects of the manuscript before submission to a peer-reviewed journal.
General structure, objectives, methodology, English grammar, the use of scientific references and the format of the figures and tables are key aspects to be revised.

---

## Round 0.2 · accepted · Accept

· Academic Editor

Accept

Authors have addressed all concerns raised by referees so it may be accepted now.